# Gene Expression Analysis of Aggressive Clinical T1 Stage Clear Cell Renal Cell Carcinoma for Identifying Potential Diagnostic and Prognostic Biomarkers

**DOI:** 10.3390/cancers12010222

**Published:** 2020-01-16

**Authors:** Jee Soo Park, Phillip M. Pierorazio, Ji Hyun Lee, Hyo Jung Lee, Young Soun Lim, Won Sik Jang, Jongchan Kim, Seung Hwan Lee, Koon Ho Rha, Nam Hoon Cho, Won Sik Ham

**Affiliations:** 1Department of Urology and Urological Science Institute, Yonsei University College of Medicine, Seoul 03722, Korea; sampark@yuhs.ac (J.S.P.); q8341@yuhs.ac (H.J.L.); silverrain@yuhs.ac (Y.S.L.); sindakjang@yuhs.ac (W.S.J.); lumpakcef@yuhs.ac (J.K.); leeseh@yuhs.ac (S.H.L.); khrha@yuhs.ac (K.H.R.); 2The James Buchanan Brady Urological Institute and Department of Urology, Johns Hopkins University School of Medicine, 600 North Wolfe Street, Park 213, Baltimore, MD 21287, USA; philpierorazio@jhmi.edu; 3Department of Oncology, Sydney Kimmel Comprehensive Cancer Center, Johns Hopkins University of Medicine, Baltimore, MD 21287, USA; 4Department of Clinical Pharmacology and Therapeutics, College of Medicine, Kyung Hee University, Seoul 02447, Korea; hyunihyuni@gmail.com; 5Department of Biomedical Science and Technology, Kyung Hee Medical Science Research Institute, Kyung Hee University, Seoul 02447, Korea; 6Department of Pathology, Yonsei University College of Medicine, Seoul 03722, Korea; cho1988@yuhs.ac

**Keywords:** clear cell renal cell carcinomas, RNA sequencing, gene expression analysis, prognostic biomarkers

## Abstract

The molecular characteristics of early-stage clear cell renal cell carcinomas (ccRCCs) measuring ≤7 cm associated with poor clinical outcomes remain poorly understood. Here, we sought to validate genes associated with ccRCC progression and identify candidate genes to predict ccRCC aggressiveness. From among 1069 nephrectomies performed on patients, RNA sequencing was performed for 12 ccRCC patients with aggressive characteristics and matched pairs of 12 ccRCC patients without aggressive characteristics. Using a prospective cohort (ClinicalTrials.gov Identifier: NCT03694912), the expression levels of nine genes (*PBRM1*, *BAP1*, *SETD2*, *KDM5C*, *FOXC2*, *CLIP4*, *AQP1*, *DDX11*, and *BAIAP2L1*) were measured by reverse-transcription polymerase chain reaction from frozen tissues, and their relation to Fuhrman grade was investigated in 70 patients with small ccRCC (≤4 cm). In total, 251 genes were differentially expressed and presented fold changes with *p*-values < 0.05; moreover, 10 genes with the greatest upregulation or downregulation in aggressive ccRCC remained significant even after adjustment. We validated previously identified genes that were associated with ccRCC progression and identified new candidate genes that reflected the aggressiveness of ccRCC. Our study provides new insight into the tumor biology of ccRCC and will help stratify patients with early-stage ccRCC by molecular subtyping.

## 1. Introduction

Approximately one-third of patients treated for localized clear cell renal cell carcinoma (ccRCC), the most common subtype of RCC, relapse following surgery; moreover, 15% of these patients exhibit metastatic potential, which can lead to death [1,2,3]. Tumor growth and changes in radiographic images are time-dependent predictors of poor oncological outcomes, whereas tumor grade, tumor necrosis, and patient performance are subject to inter-observer variability [1,2,3,4]. Alternatively, multigene assays may provide prognostic information beyond what is possible with traditional approaches and are now included in standard treatment guidelines for some tumors [4].

Recently, three companion articles were published based on multicenter, prospective longitudinal cohort studies on ccRCC patients, referred to as TRACERx Renal (tracking renal cell cancer evolution through therapy). They provide insights into ccRCC, including its origin of complex genetic alterations, evolution, and progression to metastasis [5,6,7,8]. More than 90% of ccRCC cases show inactivation of the gene encoding the von Hippel-Lindau tumor suppressor on chromosome 3p. Additionally, mutations in other tumor suppressor genes are associated with ccRCC tumorigenesis, including polybromo 1 (*PBRM1*), BRCA1-associated 1 (*BAP1*), and SET domain-containing 2 (*SETD2*). *PBRM1* and *BAP1* mutations are largely mutually exclusive, and *BAP1* mutations are significantly associated with high-grade, high-stage tumors that result in low survival [9]. Although mutations in lysine-specific demethylase 5C (*KDM5C*) occur at a lower frequency, they are also associated with poor oncological outcomes [10].

Forkhead box protein C2 (*FOXC2*) and cytoskeleton-associated protein-glycine (CAP-Gly) rich domain-containing linker protein family member 4 (*CLIP4*) mutations were associated with early-stage ccRCC and synchronous metastasis [3]. Furthermore, *PBRM1*, *BAP1*, and *FOXC2* were shown to be significantly associated with aggressive early-stage ccRCC through target sequencing and immunohistochemistry in our previous study [11].

Presently, molecular profiling does not influence the decision-making processes for ccRCC, and RNA sequencing (RNA-seq) data for aggressive early-stage ccRCC patients have not been reported. Moreover, the association between candidate gene mutations and aggressiveness has also not been reported. Hence, this study aims to identify the biomarkers associated with aggressive early-stage ccRCC. 

## 2. Results

### 2.1. Baseline Characteristics

Clinicopathologic characteristics of the patients included in this study (n = 24) have been summarized in Appendix A, and specific patient information is listed in Table 1. The mean tumor size was 4.9 ± 1.6 cm. Patients with aggressive ccRCC had no significant differences in age, BMI, tumor size, Fuhrman grade, or invasion status when compared with those with non-aggressive ccRCC. Among the 24 patients, six patients had synchronous metastasis to the lung (57.1%) and bone (42.9%). Seven patients reported tumor recurrence after nephrectomy within a mean follow-up period of 25.3 months. Cancer-specific death was reported for eight patients, with a mean survival time of 44.1 months.

### 2.2. Results from the RNA-Seq Analysis

RNA-seq analysis generated 8266 × 10^6^ base pairs (bp) from ccRCC with aggressive characteristics and 7583 × 10^6^ bp from ccRCC without aggressive characteristics. In total, 71.68 × 10^6^ reads in ccRCC with aggressive characteristics and 66.69 × 10^6^ reads in ccRCC without aggressive characteristics were mapped. There were no differences in the number of reads between the two groups, among which a total of 12,636 genes were identified. 

### 2.3. Differentially Expressed Genes

In total, 251 genes were differentially expressed with fold changes and *p*-values < 0.05; 153 upregulated genes and 98 downregulated genes had fold changes ≥2 in ccRCC with aggressive characteristics when compared with matched ccRCC without aggressive characteristics (Figure 1 and Appendix A). The ten genes that were most upregulated or downregulated in patients with aggressive ccRCC and retained significance even after adjustment are summarized in Table 2 and represented as box plots in Appendix A.

Four genes were upregulated: molybdenum cofactor sulfurase (*MOCOS*), RANBP2-like and GRIP domain containing 8 (*RGPD8*), brain-specific angiogenesis inhibitor 1 associated protein 2 like 1 (*BAIAP2L1*), and DEAD/H-box helicase 11 (*DDX11*). The six downregulated genes were solute carrier family 16 member 9 (*SLC16A9*), fraser extracellular matrix complex subunit 1 (*FRAS1*), natriuretic peptide receptor 3 (*NPR3*), aquaporin 1 (*AQP1*), transmembrane protein 38B (*TMEM38B*), and prune homolog 2 (*PRUNE2*). 

For the ten selected genes, a supervised hierarchical clustering analysis of patients with aggressive ccRCC versus matched pairs of patients with non-aggressive ccRCC was performed and revealed clustering of Pt1-4 and Pt1-9 in the non-aggressive ccRCC group (Appendix A).

We observed clear clusters distinguishing patients from the aggressive and non-aggressive ccRCC groups, as determined by the principal component analysis (PCA; Figure 2). 

Pt1-1, Pt1-2, and Pt1-12 (lung metastasis group) were grouped separately from other patients, while Pt1-4 and Pt1-9 (bone metastasis group) were not clearly separated from patients with non-aggressive ccRCC. The expression of ten genes was compared between the lung metastasis and bone metastasis groups and other patients. Significant overexpression of *MOCOS* and *BAIAP2L1* was observed in the lung metastasis group (*p* < 0.001 and *p* = 0.012, respectively) (Appendix A). Mutations were found in the six candidate genes (*PBRM1* (11/24, 45.8%), *BAP1* (6/24, 25.0%), *SETD2* (24/24, 100.0%), *KDM5C* (9/24, 37.5%), *FOXC2* (6/24, 25.0%), and *CLIP4* (8/24, 33.3%)) in more than 25% of the patients from our cohort; Figure 3 provides a visual depiction of the frequency of target gene mutations in our cohort. 

There were no significant differences in the frequencies of the six candidate aggressiveness-associated mutations in patients with aggressive ccRCC and those without aggressive ccRCC (Appendix A); however, for the patients with aggressive ccRCC, *BAP1*, *KDM5C*, and *FOXC2* mutations were enriched by two-fold, and *CLIP4* mutations were enriched by three-fold, compared to that in patients without aggressive ccRCC. 

We further verified the expression of 16 genes from the TCGA ccRCC database using UALCAN (Appendix A). Our results showed that 15 of the 16 genes (all except *RGPD8*) were differentially expressed in ccRCC samples, compared to the normal kidney samples. Moreover, the expression levels of 13 out of 16 genes (except *RGPD8*, *NPR3*, and *KDM5C*) differed according to the cancer stage.

### 2.4. Association between Oncological Outcomes and Expression of the 10 Newly Selected Genes

Cancer-specific survival (CSS) and recurrence-free survival (RFS) and their association with the ten newly selected genes were analyzed. For CSS, patients with cancer-specific death presented higher expression patterns of *RGPD8*, *BAIAP2L1*, and *DDX11* and lower expression patterns of *SLC16A9*, *FRAS1*, *AQP1*, *TMEM38B*, and *PRUNE2*, in both the univariate and multivariate analyses. For RFS, those with cancer recurrence presented higher expression patterns of *DDX11* and lower expressions of *TMEM38B* and *PRUNE2*, in both the univariate and multivariate analyses. Genes that were significantly associated with both CSS and RFS included *DDX11*, *TMEM38B*, and *PRUNE2* (Table 3). 

We have dichotomized the expression of *DDX11*, *TMEM38B*, and *PRUNE2* as negative vs. positive and evaluated their association with CSS and RFS. Patients who had *DDX11*-positive and *TMEM38B*-negative tumors had significantly higher chances of cancer-specific death (odds ratio [OR] 49.000; 95% confidence interval [CI] 3.765–637.794) and recurrence (OR 28.000, 95% CI 2.399–326.735; Table 4 and Appendix A). 

Patients with *DDX11*-positive and *TMEM38B*-negative tumors had a median CSS of 42.5 months (95% CI 28.3–56.7 months), which was substantially shorter than that of patients who had *DDX11*-negative or *TMEM38B*-positive tumors, whose median CSS was 96.4 months (95% CI 76.0–96.8 months). The differences in CSS corresponded to a hazard ratio (HR) of 16.057 (95% CI 1.866–138.169, *p* = 0.011; Figure 4). For RFS, patients with *DDX11*-positive and *TMEM38B*-negative tumors had a median RFS of 27.3 months (95% CI 15.4–39.2 months), which was substantially shorter than that of patients with *DDX11*-negative or *TMEM38B*-positive tumors, who presented a median RFS of 84.9 months (95% CI 71.9–97.8 months). The differences in RFS corresponded to an HR of 23.052 (95% CI 2.686–197.844, *p* = 0.004; Figure 4).

### 2.5. Survival Analysis of 16 Genes

Using the Kaplan–Meier plotter database, the prognostic value of the 16 genes was evaluated in 530 patients with ccRCC. We found that 13 of the 16 genes (*MOCOS*, *BAIAP2L1*, *DDX11*, *CLIP4*, *SLC16A9*, *FRAS1*, *NPR3*, *AQP1*, *PRUNE2*, *TMEM38B*, *PBRM1*, *BAP1*, and *SETD2*) were related to overall survival (OS) in patients with ccRCC (Appendix A). High expression of *MOCOS*, *BAIAP2L1*, *DDX11*, and *CLIP4*, and low expression of *SLC16A9*, *FRAS1*, *NPR3*, *AQP1*, *PRUNE2*, *TMEM38B*, *PBRM1*, *BAP1*, and *SETD2*, was significantly associated with poor OS (*p* < 0.05 for both). Expression of *RGPD8*, *KDM5C*, and *FOXC2* was not significantly associated with poor OS in patients with ccRCC.

### 2.6. GO Analysis and KEGG Analysis of DEGs

DEGs were functionally classified into biological process (BP), cellular component (CC), and molecular function (MF) categories (Appendix A). In the BP category, the top three most enriched terms were “single-organism process,” “cellular process,” and “single-organism cellular process” (Appendix A). In the CC category, the top three most enriched terms were “cell part,” “cell,” and “organelle” (Appendix A). In the MF category, the top three most enriched terms were “binding,” “protein binding,” and “ion binding” (Appendix A). Moreover, the top three most enriched terms in the KEGG analysis were “metabolic pathways,” “cell cycle,” and “complement and coagulation cascades” (Appendix A).

### 2.7. Validation of Target Genes Using Frozen Tissue PCR

Of the ten genes newly identified through RNA-seq analysis, *AQP1*, *DDX11*, and *BAIAP2L1* were previously reported to be potent indicators of outcome in patients with ccRCC. We analyzed the expression of these three genes along with that of the six genes previously identified by the qRT-PCR analysis of frozen tissue samples. Our results showed that *DDX11* expression and tumor size were significantly greater in patients with high Fuhrman grade (3 and 4), both in the univariate and multivariate analyses (Appendix A).

## 3. Discussion

This is the first study to identify 251 DEGs in early-stage aggressive ccRCCs measuring ≤7 cm using RNA-seq. Among the potential target candidate genes, *BAP1*, *KDM5C*, *FOXC2*, and *CLIP4* mutations were enriched in patients with aggressive ccRCC. Among the DEGs identified in this study, *MOCOS*, *RGPD8*, *BAIAP2L1*, and *DDX11* were significantly upregulated; in contrast, S*LC16A9*, *FRAS1*, *NRP3*, *AQP1*, *TMEM38B*, and *PRUNE2* were significantly downregulated. 

A panel of six genes (*PBRM1*, *BAP1*, *SETD2*, *KDM5C*, *FOXC2*, and *CLIP4*) was chosen for mutational frequency analysis based on the clinical relevance of significantly mutated genes across all ccRCC cohorts [12,13]. However, these six genes were not differentially expressed, perhaps because our patient cohort included unique cases for which cancers exhibited aggressive characteristics, such as synchronous metastasis, recurrence, or cancer-specific death, even though the clinical stage was low. Moreover, small sample sizes would have affected this result. As in other large NGS studies, *PBRM1* mutation was identified in 45.8% patients. However, *BAP1* mutation was observed in 25% of patients, which is slightly higher than the value obtained in other studies, perhaps because our cohort comprised patients with more aggressive ccRCC; we intentionally collected aggressive ccRCC to pair with non-aggressive ccRCC. *BAP1*, a critical gatekeeper for disease progression, has a mutually exclusive interaction with *PBRM1*, whereas the *BAP1* mutation is associated with worse prognosis and a higher Fuhrman grade than the *PBRM1* mutation [7,14]. This finding was further supported by the MSKCC cohort study where *BAP1* mutations were associated with poor prognostic factors (higher T stage, higher nuclear grade, large size, more necrosis), and the presence of metastatic disease at presentation) [15].

*SETD2* mutation was observed in all cases of ccRCC in our study, which diverges from other NGS studies in which the frequency of *SETD2* mutation was substantially lower (approximately 11%) [12,13]. However, Liu et al. reported a multicenter study in China that had evaluated radical nephrectomy cases, wherein the SETD2 protein deficiency rate was 34.1%, although there may have been a discrepancy between protein deficiency and gene mutation rates [16]. *KDM5C* mutations were found in less than 40% of the study population, compared to 3.8–6.8% reported in other NGS studies [12,13]. We believe that this difference in *KDM5C* mutation frequency could result from selection bias because of the unique group of patients who were defined as aggressive. However, owing to the intratumoral heterogeneity (ITH), there could also have been a sampling bias. Moreover, gene mutation status was based on multiple types of sequencing that only reflect the gene status of very circumscribed cancer lesions, and thus could underestimate mutational rates [16].

The frequency of *PBRM1* mutation was not different in ccRCC with and without aggressive characteristics. *BAP1*, *KDM5C*, *FOXC2*, and *CLIP4* mutations were enriched in aggressive ccRCC, which was not statistically significant because of small sample sizes. This finding is similar to the results of other studies that have reported an association between *BAP1* and *KDM5C* mutations and aggressiveness of tumors [9,10]. *FOXC2*, a gene known to be associated with tumor aggressiveness and synchronous metastasis in ccRCC, based on our previous studies [2,11], was found to be associated with aggressive characteristics of tumors in early-stage ccRCC, in the present study. The frequency of *PBRM1* mutation was similar between patients with aggressive and non-aggressive ccRCC, perhaps because it is an early, essential event in tumorigenesis that does not impact clinical outcome, but instead plays a principal role in tumor initiation. *BAP1* mutation was more frequently observed in aggressive ccRCC; *BAP1* is believed to be associated with disease progression and a worse prognosis [17].

With regard to metastasis, the lung and bone were the two most common sites, with up to 60% and 40% of patients having lung and bone metastasis, respectively [18]. A recent study suggests that patients with bone metastases have an unfavorable prognosis; however, another study found no prognostic differences between metastatic sites [19], although this remains controversial. According to this study, synchronous bone metastatic patients were grouped with non-aggressive ccRCC patients, while synchronous lung metastatic patients were grouped with the aggressive ccRCC group. These results show that the early bone metastasis group genetically clusters with the less aggressive ccRCC group and suggest that the prognosis of early bone metastasis ccRCC patients might not be bad, although future studies are needed to verify this finding. The exact role of *MOCOS* and *BAIAP2L1* in lung metastasis needs to be studied further, as these genes may help us understand differences in metastatic potential in ccRCC and the prognostic value of different metastatic sites. 

We included *AQP1*, *DDX11*, and *BAIAP2L1* from the newly identified gene set because these were previously associated with ccRCC or have a known relation to other tumors [20,21,22]. A prospective study examining frozen tissue qRT-PCR of *AQP1*, *DDX11*, *BAIAP2L1*, and six previously identified genes (*PBRM1*, *BAP1*, *SETD2*, *KDM5C*, *FOXC2*, and *CLIP4*) showed that expression of *DDX11* was a significant factor for predicting tumor aggressiveness based on Fuhrman grade. Bhattacharya et al. demonstrated that inhibiting *DDX11* in melanoma cells decreased proliferation and rapidly increased apoptosis [21]. In the future, we will perform an in vitro and in vivo study using *DDX11* since ccRCC has similar tumor characteristics as melanoma.

We only included *DDX11* and *TMEM38B* in our final analysis of CSS and RFS since although using three genes increased the statistical significance (Appendix A), the combination of *DDX11* and *PRUNE2* was not associated with recurrence in the multivariate analysis (Table 4). Therefore, although combining three genes would of course increase the statistical significance compared to two genes in clinical settings, it is also important to reduce the number of genes used. Therefore, we tried to select the most important gene that is both related to CSF and RFS.

We propose that the factors involved in tumor aggressiveness and metastasis or recurrence are different. Previous studies have focused on using Fuhrman grade as the golden standard and have compared gene expression or mutation based on Fuhrman grade. However, Fuhrman grade only reflects cell differentiation and does not reflect metastatic and recurrence potential. Recent studies have reported clonal and subclonal mutations in ccRCC, whereas clonal mutations are present in all tumor regions, and subclonal mutations are present in some, but not all, tumors [23]. Clonal cancer evolution is also known to be involved in ccRCC through Darwinian selection [23]. TRACERx Renal classified tumors into seven distinct evolutionary subtypes [6,7], and primary tumors with low ITH, in addition to a low fraction of the tumor genome affected by somatic copy-number alterations (SCNAs), had an overall low metastatic potential. In contrast, primary tumors with low ITH but elevated SCNAs were associated with rapid progression at multiple sites [7]. Thus, our knowledge regarding potential ccRCC biomarkers for guiding intervention and surveillance has recently been dramatically expanded. Recent article by Wach et al. reported the RNA sequencing of collecting duct carcinoma, a rare renal cell carcinoma subtype with a very poor prognosis and solute carrier (SLC) gene family (*SLC3A1*, *SLC9A3*, *SLC26A7*, and *SLC47A1*) was included in the most downregulated genes [24]. According to this study trend, our newly identified biomarkers may further aid in unraveling the evolutionary trajectories of ccRCC tumor growth. 

This study has a few limitations. First, this study intentionally collected aggressive cases of ccRCC; thus, selection bias may have resulted in statistical differences from other NGS studies. However, this collection method has benefits, including the examination of unique cases of interest to clinicians. Second, because of the retrospective nature of this study, the study sample size was small; this limited our ability to further investigate the ITH of primary tumors, although the ITH in ccRCC tumors, including even small renal masses, may be substantial [1]. In future studies, we plan to overcome this ITH limitation by analyzing the circulating tumor DNA in plasma. Third, RNA-seq was based only on surgical FFPE samples and we could not provide validation results from a different independent dataset. To overcome these limitations and validate our findings, a prospective clinical trial is now underway (ClinicalTrials.gov Identifier: NCT03694912) that will include an examination of the mutation profiles of the selected genes (*PBRM1*, *BAP1*, *SETD2*, *KDM5C*, *FOXC2*, *CLIP4*, *MOCOS*, *BAIAP2L1*, *DDX11*, and *AQP1*) for aggressive ccRCC. However, our preliminary data obtained using frozen tissue PCR show promising results for *DDX11*, and in vivo and in vitro studies on each gene are ongoing. The demonstration of immunohistochemistry in selected genes would have provided the association between RNA and protein. However, we have focused on the use of RNA expression profiles since it is easier and faster to use in the clinical settings compared to immunohistochemistry. Nevertheless, the protein expression profiles would provide more specific information regarding the role of selected genes. Therefore, we are planning to include protein expression analysis in the future study.

## 4. Materials and Methods

### 4.1. Patients and Tissues

For this retrospective study, data were gathered from 1132 patients with ccRCC (≤7 cm) who underwent radical and partial nephrectomy between January 2008 and December 2014. An aggressive tumor was defined as a tumor exhibiting synchronous metastasis, recurrence, or cancer-specific death; synchronous metastasis was defined as metastasis detected at or within three months of the primary RCC diagnosis [25]. Twelve patients with aggressive ccRCC and 12 matched patients in terms of gender, BMI, percentage of radical surgeries performed, clinical tumor sizes, and Fuhrman grades with non-aggressive ccRCC were examined in order to adjust the confounding factors other than aggressiveness of ccRCC. This study was approved by the Institutional Review Board of the Yonsei University Health System (project no: 4-2013-0742).

We included patients with ccRCC (≤7 cm) treated with nephrectomy alone and for whom formalin-fixed paraffin-embedded (FFPE) tumor tissue samples were available; we also included patients who did not exhibit the typical characteristics of ccRCC. Specimens that were inappropriate for molecular analysis were excluded, and clinicopathologic features were recorded for each patient (Appendix A). Diameters of the primary tumors were measured using imaging modalities. 

For validation analyses, we used a prospective cohort (ClinicalTrials.gov Identifier: NCT03694912) sampled between November 2018 and July 2019 and included cases with small ccRCCs (≤4 cm). Data on clinicopathologic features were included, as previously described in our retrospective study group. The levels of six previously identified genes (*PBRM1*, *BAP1*, *SETD2*, *KDM5C*, *FOXC2*, and *CLIP4*), plus three genes (*AQP1*, *DDX11*, and *BAIAP2L1*) among the ten newly identified genes, were considered potent indicators of outcome in ccRCC patients according to a literature search. Gene expression was measured by reverse-transcription polymerase chain reaction (qRT-PCR).

### 4.2. Tissue Preparation

Formalin-fixed, paraffin-embedded (FFPE) sections from patients with ccRCC were obtained from the archives of the Department of Pathology at Yonsei University College of Medicine (Seoul, Korea). All cases were reviewed and classified by a urologic pathologist (N.H.C.). Non-tumor elements were identified based on hematoxylin and eosin-stained slides from each sample by a urologic pathologist (N.H.C.). The samples were subsequently cut into 20-μm sections before being transferred to an extraction tube.

### 4.3. RNA Extraction and Sequencing

Briefly, for FFPE tissues, RNA was extracted from 20 μm sections using an FFPE RNeasy kit (Qiagen, Gaithersburg, MD, USA), and 100 ng of RNA was used to construct a sequencing library using a TruSeq RNA Access library prep kit (Illumina, San Diego, CA, USA) according to the manufacturer’s protocols (Appendix A).

The sequencing libraries were quantified by the Kapa Biosystems Library Quantification Kit for Illumina Sequencing platforms according to the qPCR Quantification Protocol Guide (KapaBiosystems, Wilmington, MA, USA Cat #KK4854) and qualified by the TapeStation D1000 ScreenTape (Agilent Technologies, Palo Alto, CA, USA Cat # 5067-5582). Indexed libraries were then submitted to Illumina NovaSeq 6000 (Illumina), and paired-end (2 × 101 bp) sequencing was performed by Macrogen Incorporated.

### 4.4. Analysis of RNA Sequencing—Differentially Expressed Gene (DEG) Selection

Paired-end sequencing reads from cDNA libraries (101 bp) were generated and the trimmed reads were aligned to the reference human genome ([University of California Santa Cruz] UCSC hg19). Transcriptome assembly of known transcripts, novel transcripts, and alternative splicing transcripts was processed.

Based on the results of this processing, the abundance of transcript and gene expression was calculated as read count (the number of reads mapping to a gene) or FPKM value (fragments per kilobase of exon per million fragments mapped) per sample. To identify DEGs from aggressive and non-aggressive ccRCC groups, genes with more than one sample of zero read count were pre-filtered. We then measured the expression levels of 27,685 RefSeq genes and 12,636 genes (Appendix A).

For DEGs, unsupervised hierarchical clustering analysis was performed with the complete linkage method and Euclidean distance as a measure of similarity. Principal component analysis (PCA) was performed to reduce the dimensionality of the data set by transforming it into a new set of variables to summarize the data features process (Appendix A). All data analysis and visualization of DEGs were conducted using R.3.5.1 (www.r-project.org).

### 4.5. Variant Calling

For variant calling of RNA-seq data, trimmed reads were aligned to the human genome (UCSC hg19). The alignment of the reads used in this analysis was created through Split ‘N’ Trim, mapping quality reassignment, indel realignment, and base recalibration process (Appendix A). For our target gene list (*PBRM1*, *BAP1*, *SETD2*, *KDM5C*, *FOXC2*, and *CLIP4*), the frequency of alterations in all samples was represented on the heatmap. Individual genes are represented as rows, and individual patients are represented as columns.

### 4.6. qRT-PCR

Total RNA was extracted from frozen tissue samples using TRIzol^®^ reagent (Ambion, Life Technologies, Carlsbad, CA, USA), and 1 μg of total RNA was reverse-transcribed into first-strand cDNA using an iNtRon Maxime RT PreMix (Intronbio, Sungnam, Korea Cat No. 25081) according to the manufacturer’s protocol (Appendix A). 

### 4.7. GO Analysis and KEGG Analysis

Gene Ontology (GO) analysis per biological process (BP), cellular component (CC), and molecular function (MF) for DEGs were performed based on Gene Ontology (http://geneontology.org/). KEGG (Kyoto Encyclopedia of Genes and Genomes) pathway analysis for DEGs was also performed based on the KEGG pathway (https://www.genome.jp/kegg/) database (Appendix A).

### 4.8. Validation of Gene Expression, Survival Analyses, and Statistical Analyses

UALCAN (http://ualcan.path.uab.edu) was used to validate the expression of ten newly identified genes and six target genes according to the stage of the cancer. The Kaplan–Meier plotter database (http://kmplot.com) was used to analyze the associations between 16 genes (10 newly identified and six target genes) and overall patient survival. Additional details for statistical analyses are provided in Appendix A.

## 5. Conclusions

In summary, we identified DEGs in aggressive early-stage ccRCC, defined as those with synchronous metastasis, recurrence, or cancer-specific death, and found that *MOCOS*, *RGPD8*, *BAIAP2L1*, and *DDX11* showed significantly higher expression levels, whereas *SLC16A9*, *FRAS1*, *AQP1*, *TMEM38B*, and *PRUNE2* showed significantly lower expression levels compared to those in non-aggressive ccRCC. Moreover, we verified previously known genes that are associated with tumor aggressiveness by analyzing mutational frequency. Further research on these molecular biomarkers may help stratify patients with early-stage ccRCC by molecular subtyping and aid in making clinical decisions accordingly.

## Figures and Tables

**Figure 1 cancers-12-00222-f001:**
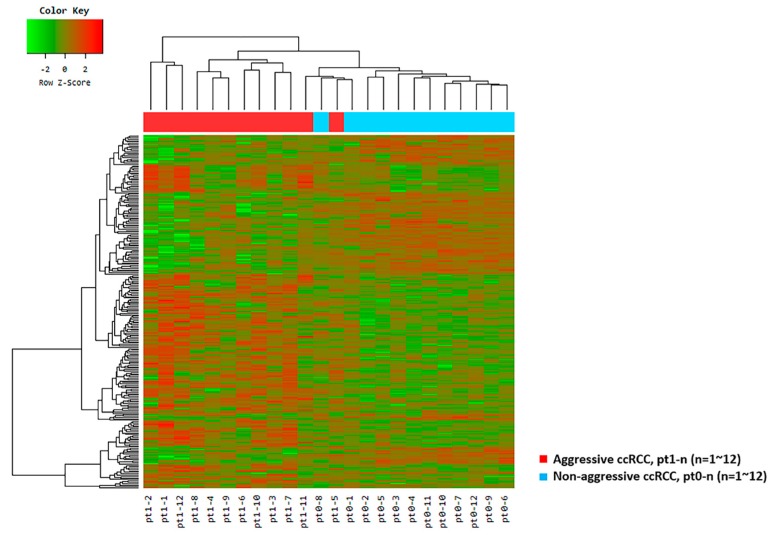
Unsupervised hierarchical clustering analysis (red, high relative expression; green, low relative expression) in aggressive clear cell renal cell carcinomas (ccRCC) patients (n = 12, red) versus matched pairs of non-aggressive ccRCC patients (n = 12, blue) based on 251 differentially expressed genes.

**Figure 2 cancers-12-00222-f002:**
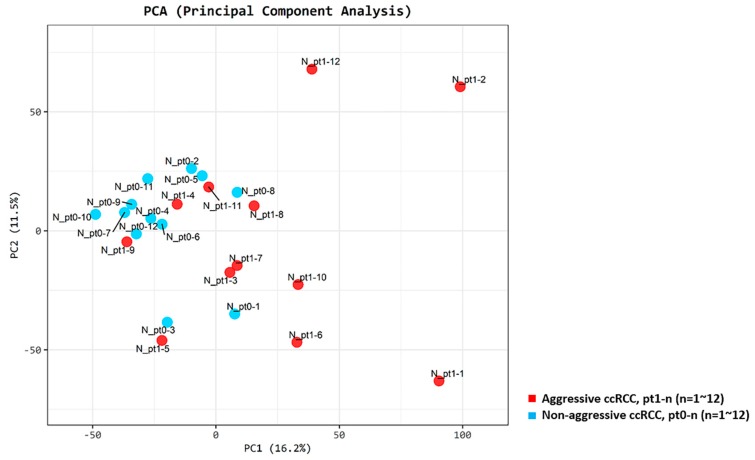
Principal component score plots. First and second principal component scores in patients with aggressive ccRCC (n = 12, red) versus matched patient pairs with non-aggressive ccRCC (n = 12, blue) are plotted.

**Figure 3 cancers-12-00222-f003:**
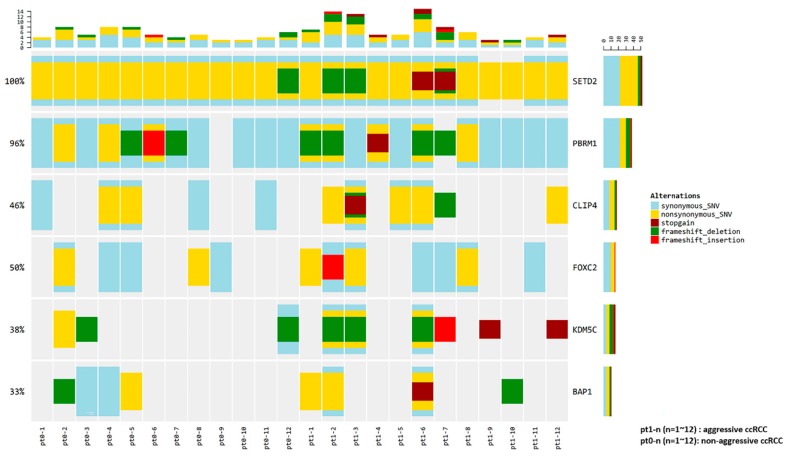
Frequency of candidate gene mutations in the patient cohort, and comparison of those in aggressive and non-aggressive ccRCC.

**Figure 4 cancers-12-00222-f004:**
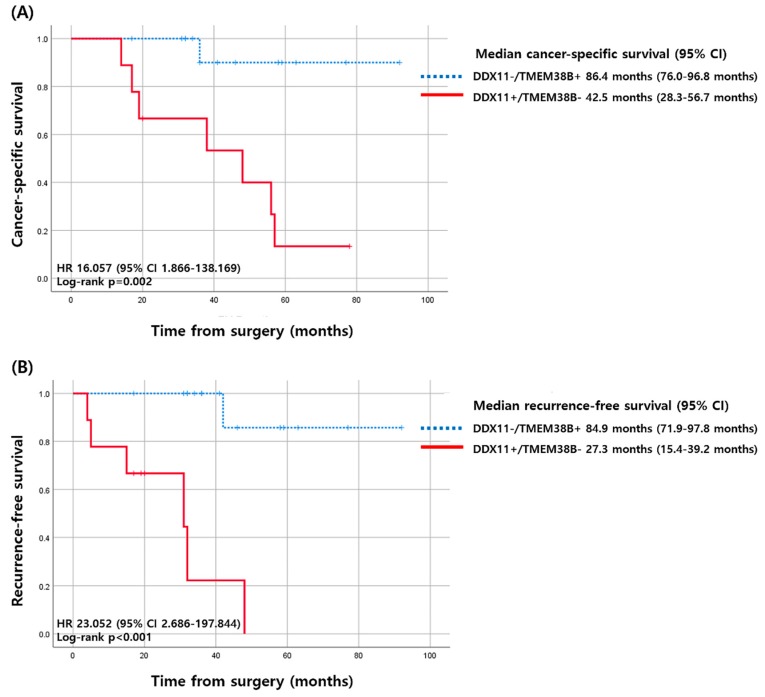
Kaplan–Meier curves of cancer-specific survival and recurrence-free survival according to the expression of *DDX11* and *TMEM38B*. (**A**) Cancer-specific survival (**B**) recurrence-free survival. CI, confidence interval; HR, hazard ratio.

**Table 1 cancers-12-00222-t001:** Clinicopathological characterization of each patient.

Aggressiveness	Patient ID	Sex	Age (yr)	Tumor Size (cm)	Fuhrman Grade	Invasion (Perinephric/Sinus Fat/Vascular)	Survival Time ^a^ (m)	Outcome ^b^	Recurrence Site	Synchronous Metastatic Site
N	Pt0-1	M	67	1.6	3	N	41	ned		
N	Pt0-2	M	56	6.3	3	Y	36	ned		
N	Pt0-3	M	76	3.5	2	Y	32	ned		
N	Pt0-4	F	73	5.3	2	Y	92	ned		
N	Pt0-5	M	81	6.6	3	Y	17	ned		
N	Pt0-6	M	72	5.4	3	Y	77	ned		
N	Pt0-7	M	60	6.9	3	Y	59	ned		
N	Pt0-8	M	59	6.4	3	Y	63	ned		
N	Pt0-9	M	61	4.7	3	Y	58	ned		
N	Pt0-10	M	76	5.0	2	N	46	ned		
N	Pt0-11	M	65	4.4	2	N	32	ned		
N	Pt0-12	M	75	2.7	2	N	34	ned		
Y	Pt1-1	M	73	5.5	3	N	19	cd		Lung
Y	Pt1-2	M	73	6.9	4	Y	17	cd		Lung
Y	Pt1-3	M	60	5.1	3	Y	57	pd/cd	Local recurrence, liver	
Y	Pt1-4	M	58	2.8	3	N	36	cd		Bone
Y	Pt1-5	M	74	1.6	2	N	48	pd/cd	Lung, lymph nodes	
Y	Pt1-6	M	70	6.6	3	Y	38	pd/cd	Liver, bone, lymph nodes	
Y	Pt1-7	M	61	6.7	3	Y	14	pd/cd	Lung, liver	Lung, bone
Y	Pt1-8	M	54	4.8	2	Y	56	pd/cd	Local recurrence	
Y	Pt1-9	M	66	4.2	2	N	20	ned		Bone
Y	Pt1-10	M	60	5.2	4	Y	78	pd	Lung, bone	
Y	Pt1-11	M	70	3.5	2	N	58	pd	Bone	
Y	Pt1-12	M	44	5.5	3	Y	31	ned		Lung

^a^ Survival time was defined as the time from nephrectomy until the patient’s death or the last time that the patient was known to be alive; ^b^ cd, cancer death; ned, no evidence of disease; pd, progression of the disease.

**Table 2 cancers-12-00222-t002:** Significantly upregulated/downregulated genes in clinical T1 stage with or without aggressive characteristics (metastasis, recurrence, or cancer-specific death).

*Upregulated*						
Gene Symbol	Gene Title	FPKM (Mean ± SD)		Log_2_ Fold Change	*p*-Value ^a^	Adjusted *p* ^b^
		RCC with Aggressive Characteristics	RCC without Aggressive Characteristics	(With Aggressive/without Aggressive Characteristics)		
MOCOS	Molybdenum cofactor sulfurase	16.76 ± 15.41	1.40 ± 1.29	+3.77	9.29 × 10^−6^	1.66 × 10^−2^
RGPD8	RANBP2-like and GRIP domain containing 8	41.49 ± 35.62	6.51 ± 5.38	+2.78	3.67 × 10^−5^	4.59 × 10^−2^
BAIAP2L1	BAI1 associated protein 2 like 1	7.05 ± 3.39	2.18 ± 1.28	+1.87	2.18 × 10^−6^	7.98 × 10^−3^
DDX11	DEAD/H-box helicase 11	31.95 ± 13.09	14.25 ± 5.18	+1.32	2.55 × 10^−6^	7.98 × 10^−3^
***Downregulated***						
**Gene Symbol**	**Gene Title**	**FPKM (Mean ± SD)**		**Log_2_ Fold Change**	***p*-Value ^a^**	**Adjusted *p*^b^**
		**RCC with Aggressive Characteristics**	**RCC without Aggressive Characteristics**	**(With Aggressive/without Aggressive Characteristics)**		
SLC16A9	Solute carrier family 16 member 9	2.13 ± 2.51	11.90 ± 8.48	−2.41	5.91 × 10^−6^	1.48 × 10^−2^
FRAS1	Fraser extracellular matrix complex subunit 1	3.15 ± 3.80	13.66 ± 6.60	−2.12	4.37 × 10^−8^	5.47 × 10^−4^
NPR3	Natriuretic peptide receptor 3	3.22 ± 3.37	12.53 ± 7.84	−1.88	3.61 × 10^−5^	4.59 × 10^−2^
AQP1	Aquaporin 1 (Colton blood group)	11.02 ± 11.86	36.80 ± 17.38	−1.63	1.73 × 10^−5^	2.71 × 10^−2^
TMEM38B	Transmembrane protein 38B	2.74 ± 1.81	8.07 ± 2.70	−1.47	1.21 × 10^−6^	7.57 × 10^−3^
PRUNE2	Prune homolog 2	19.10 ± 9.42	49.39 ± 15.65	−1.24	8.48 × 10^−6^	1.66 × 10^−2^

FPKM, fragments per kilobase of exon per million fragments mapped; RCC, renal cell carcinoma; SD, standard deviation; ^a^
*p*-values by nbiomTest function in DESeq2; ^b^ Adjusted *p*-values by the Benjamini–Hochberg algorithm.

**Table 3 cancers-12-00222-t003:** Comparison of target gene expression according to oncological outcomes (cancer-specific death and recurrence).

	Cancer-Specific Death				
FPKM (Mean ± SD)	RCC with Cancer-Specific Death (n = 8)	RCC without Cancer-Specific Death (n = 16)	*p*-Value ^a^	Multivariate OR (95% CI)	*p*-Value ^b^
**MOCOS**	16.4 ± 12.5	5.4 ± 12.4	0.054	-	0.093
**RGPD8**	47.6 ± 38.4	12.2 ± 17.5	0.036	1.048 (1.006–1.093)	0.026
**BAIAP2L1**	7.5 ± 4.0	3.2 ± 2.2	0.019	1.553 (1.088–2.215)	0.015
**DDX11**	34.3 ± 13.4	17.5 ± 9.3	0.002	1.130 (1.025–1.245)	0.014
**SLC16A9**	1.3 ± 2.3	9.9 ± 8.2	0.001	0.585 (0.362–0.946)	0.029
**FRAS1**	2.4 ± 2.5	11.4 ± 7.4	<0.001	0.741 (0.567–0.967)	0.028
**NPR3**	3.1 ± 3.4	10.3 ± 8.0	0.006	-	0.058
**AQP1**	11.3 ± 12.8	30.2 ± 19.7	0.023	0.926 (0.859–0.997)	0.042
**TMEM38B**	1.9 ± 1.2	7.2 ± 2.9	<0.001	0.277 (0.091–0.841)	0.024
**PRUNE2**	20.1 ± 8.5	41.3 ± 20.5	0.002	0.921 (0.854–0.993)	0.032
	**Recurrence**				
**FPKM (Mean ± SD)**	**RCC with Recurrence (n = 7)**	**RCC without Recurrence (n = 17)**	***p*-Value ^a^**	**Multivariate OR (95% CI)**	***p*-Value ^b^**
**MOCOS**	9.5 ± 6.6	8.9 ± 15.4	0.921	-	0.917
**RGPD8**	65.5 ± 25.9	6.9 ± 6.0	0.001	-	0.996
**BAIAP2L1**	6.8 ± 3.0	3.7 ± 3.4	0.048	-	0.066
**DDX11**	36.0 ± 10.2	17.8 ± 10.6	0.001	1.140 (1.029–1.264)	0.012
**SLC16A9**	1.5 ± 2.1	9.3 ± 8.3	0.002	-	0.056
**FRAS1**	3.1 ± 2.8	10.6 ± 7.8	0.002	-	0.051
**NPR3**	2.9 ± 2.5	9.9 ± 8.1	0.004	-	0.073
**AQP1**	12.7 ± 13.0	28.5 ± 20.3	0.071	-	0.090
**TMEM38B**	2.5 ± 1.5	6.6 ± 3.4	0.001	0.579 (0.355–0.944)	0.028
**PRUNE2**	14.5 ± 7.9	42.4 ± 17.6	<0.001	0.788 (0.623–0.996)	0.047

Data are shown as mean ± SD; ^a^
*p*-value calculated using the *t*-test; ^b^
*p*-value calculated using logistic regression for multivariate analysis.

**Table 4 cancers-12-00222-t004:** Expression of DDX11, TMEM38B, and PRUNE2 by oncological outcomes (cancer-specific death and recurrence).

Groups Dichotomized by the Expression of DDX1, TMEM38B and PRUNE2	Cancer-Specific Death				
	RCC with Cancer-Specific Death (n = 8)	RCC without Cancer-Specific Death (n = 16)	*p*-Value ^a^	Multivariate OR (95% CI)	*p*-Value ^b^
**DDX11+**	7/8 (87.5%)	5/16 (31.3%)	0.027	-	0.065
**TMEM38B−**	8/8 (100.0%)	4/16 (25.0%)	0.001	-	0.998
**PRUNE2−**	7/8 (87.5%)	5/16 (31.3%)	0.027	-	0.065
**DDX11+ and TMEM38B−**	7/8 (87.5%)	2/16 (12.5%)	0.001	49.000 (3.765–637.794)	0.003
**DDX11+ and PRUNE2−**	6/8 (75.0%)	2/16 (12.5%)	0.005	21.000 (2.372–185.930)	0.006
	**Recurrence**				
	**RCC with Recurrence (n = 7)**	**RCC without Recurrence (n = 17)**	*** p*-Value ^a^**	**Multivariate OR (95% CI)**	***p*-Value ^b^**
**DDX11+**	7/7 (100.0%)	5/17 (29.4%)	0.005	-	0.999
**TMEM38B−**	6/7 (85.7%)	6/11 (54.5%)	0.069	11.000 (1.061–114.086)	0.045
**PRUNE2−**	7/7 (100.0%)	5/17 (29.4%)	0.005	-	0.999
**DDX11+ and TMEM38B−**	6/7 (85.7%)	3/17 (17.6%)	0.004	28.000 (2.399–326.735)	0.008
**DDX11+ and PRUNE2−**	7/7 (100.0%)	1/17 (5.9%)	<0.001	-	0.998

Data are shown as the number of patients (%); ^a^
*p*-value calculated using the chi-square test or Fisher’s exact test; ^b^
*p*-value calculated using logistic regression for multivariate analysis; DDX11+, FPKM (fragments per kilobase of exon per million fragments mapped) > 20.0; TMEM38B−, FPKM < 5.0; PRUNE2−, FPKM < 32.0.

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
