# Peer review of "Gene Expression Analysis of Aggressive Clinical T1 Stage Clear Cell Renal Cell Carcinoma for Identifying Potential Diagnostic and Prognostic Biomarkers"

_cancers, 2020, doi:10.3390/cancers12010222_

Round 1
Reviewer 1 Report
In this manuscript, the authors identified a group of novel genes for predicting ccRCC aggressiveness and progression through RNA-seq coupled analysis of a large patient cohort. These genes can serve as potential diagnostic and prognostic biomarkers for stratifying patients with early-stage ccRCC by molecular subtyping. This study also provides new insights into understanding the pathogenesis and biology of ccRCC. Overall, the manuscript is well written with data clearly presented and statistics appropriately used. This reviewer has no major concerns but a few minor points for the authors to consider for incorporation into revision.
How the “matched pairs” of ccRCC patient samples (Pt0 vs. Pt1) as used for RNA-seq were defined? Were these samples collected from the same patients at different disease stages?
It would strengthen Fig. 4 by providing similar results from a different independent dataset, such as TCGA dataset, by surveying publicly available databases.
It would be necessary to examine protein expression levels of select key candidate genes identified from RNA-seq, such as DDX11 and TMEM38B, in patient samples by immunohistochemistry, which is expected to show consistent changes with RNA-seq data.
Author Response
Response to Reviewer 1 Comments
In this manuscript, the authors identified a group of novel genes for predicting ccRCC aggressiveness and progression through RNA-seq coupled analysis of a large patient cohort. These genes can serve as potential diagnostic and prognostic biomarkers for stratifying patients with early-stage ccRCC by molecular subtyping. This study also provides new insights into understanding the pathogenesis and biology of ccRCC. Overall, the manuscript is well written with data clearly presented and statistics appropriately used. This reviewer has no major concerns but a few minor points for the authors to consider for incorporation into revision.
Point 1: How the “matched pairs” of ccRCC patient samples (Pt0 vs. Pt1) as used for RNA-seq were defined? Were these samples collected from the same patients at different disease stages?
Response: Thank you for your comments. The “matched pairs” of ccRCC samples does not imply that it is from the same patients at different disease stages. We tried to match clinicopathologic characteristics such as gender, BMI, percentage of radical surgeries performed, clinical tumor sizes, and Fuhrman grades as similar as possible between Pt0 vs. Pt1 in order to compare the parameters of aggressive characteristics (synchronous metastasis, recurrence, or cancer-specific death) by adjusting other confounding factors. In response to the reviewer’s comments, we have edited the sentence in the method section in order to clarify the reviewer’s understanding (lines 317-319).
Point 2: It would strengthen Fig. 4 by providing similar results from a different independent dataset, such as TCGA dataset, by surveying publicly available databases.
Response: Thank you for your comments. We have searched publicly available databases, however unfortunately there is no datasets regarding DDX11-/TMEM38B+ or DDX11+/TMEM38B- groups. We acknowledge that limitation of our study is that we have not provided similar results from a different independent dataset. Future validation study is underway to overcome this limitation. We have added this information in the limitation section (lines 297-298).
Point 3: It would be necessary to examine protein expression levels of select key candidate genes identified from RNA-seq, such as DDX11 and TMEM38B, in patient samples by immunohistochemistry, which is expected to show consistent changes with RNA-seq data.
Response: Thank you for your comments. We agree with the reviewer that demonstration of protein expression levels of selected genes from RNA-seq would be informative and would provide the association between mRNA and protein. However, we have focused on use of RNA expressions in predicting aggressiveness of ccRCC since it is easier and faster to use in the clinical settings compared to immunohistochemistry. Nevertheless, we agree that protein expression analysis would provide insight of molecular function of the selected genes more specifically. Therefore, we are planning to include demonstration of protein expressions of selected genes in the future study. We have added the sentences in the limitation section (lines 303-308).

Reviewer 2 Report
Major concerns:
Table 2: the authors reported the statistical results of t-test, which I believe is not appropriate as they can report the p-values from DEseq2 package. I would also suggest the authors to report log2 fold-change, which is much easier for readers to see the regulation of the genes. Table 4: How did the authors categorize patients into positive and negative groups? I also expect to see the similar analysis for the combination of 3 genes. 2: the PCA does not show a good classification of patients. SM Fig.5: The authors should also include RNA-seq data of the 6 mutated genes and then compare them with the TCGA data. The authors did not explain and discuss why regulation of some genes (e.g., MOCOS, BAIAP2L1, NPR3) is not consistent with RNA-seq data. SM Fig.6: How did they categorize the patients into low or high groups based on expression levels of the 16 genes? SM Table 4: why the expression level of DDX11 is smaller in samples with higher Fuhrman grade (3 and 4)? How about the other 7 newly identified genes from RNA-seq data, I expect to see whether or not their regulation is consistent with the published qPCR results. Why the 6 previously identified genes (PBRM1, BAP1, SETD2, KDM5C, FOXC2, and CLIP4) do not show any power for predicting tumor aggressiveness?Minor concerns:
In a recent publication (doi: 10.3390/cancers12010064), the SLC gene family genes were also identified as dysregulated in collecting duct renal cell carcinoma, which I think should be discussed by the authors. Tables: the tables should be re-formatted by adjusting the size of grids, making them easier to be read. Figures: the resolution of most figures in supplementary materials is too low to be read. Data sharing: the authors should consider making their RNA-seq data accessible by others. Either publish the read counts of genes as a supplementary file or submit the data to omnibus in Pubmed.Author Response
Response to Reviewer 2 Comments
Major concerns
Point 1: Table 2: the authors reported the statistical results of t-test, which I believe is not appropriate as they can report the p-values from DEseq2 package.
Response: Thank you for your comments. We have mistakenly written as the results of t-test. P-value was generated by nbiomTest function in DESeq2 as mentioned in the Supplementary Method 3. We edited the sentence below the Table 2 that stated as if we have performed t-test.
Point 2: Table 2: I would also suggest the authors to report log2 fold-change, which is much easier for readers to see the regulation of the genes.
Response: As suggested by the reviewer, we have changed to log2 fold-change since it would be much easier for readers to see the regulation of the genes.
Point 3: Table 4: How did the authors categorize patients into positive and negative groups?
Response: As already mentioned in Supplementary Method 7, DDX11 positive tumors were categorized as the FPKM values higher than 20.0. DDX11 negative tumors were categorized as the FPKM values less than or equal to 20.0. TMEM38B positive tumors were categorized as the FPKM values higher than or equal to 5.0. TMEM38B negative tumors were categorized as the FPKM values less than 5.0. PRUNE2 positive tumors were categorized as the FPKM values higher than or equal to 32.0. PRUNE2 negative tumors were categorized as the FPKM values less than 32.0. In order to help the reader’s understanding by only reading the table, we have added the sentence below Table 4 stating how patients were categorized into positive and negative groups.
Point 4: I also expect to see the similar analysis for the combination of 3 genes.
Response: Thank you for your comments. The analysis of combination of 3 genes (DDX11+&TMEM38B-&PRUEN2-) is reported below. Although the statistical significance is higher when using 3 genes, we have not included this result in our manuscript since DDX11+/PRUNE2- group was not associated with recurrence in multivariate analysis. Although the combination of 3 genes showed good performance compared to those that used 2 genes, in clinical settings, it is also important to reduce the number of genes that used. Therefore, we tried to select the most important genes that is both related to cancer-specific death and recurrence. In response to the reviewer’s comments, we have included this result as the Supplementary Table 4 and in the discussion section (lines 267-272).
Supplementary Table 4. Expression of combination of DDX11, TMEM38B, and PRUNE2 by oncological outcomes (cancer-specific death and recurrence) (in addition to Table 4)
|
Cancer-specific death |
|
|
|
|
|
RCC with cancer-specific death (n=8) |
RCC without cancer-specific death (n=16) |
P-valuea |
Multivariate OR (95% CI) |
P-valueb |
DDX11+&TMEM38B-&PRUNE2- |
6/8 (75.0%) |
1/16 (6.2%) |
0.001 |
45.000 (3.408-594.116) |
0.004 |
|
Recurrence |
|
|
|
|
|
RCC with recurrence (n=7) |
RCC without recurrence (n=17) |
P-valuea |
Multivariate OR (95% CI) |
P-valueb |
DDX11+&TMEM38B-&PRUNE2- |
6/7 (85.7%) |
1/17 (5.9%) |
<0.001 |
96.000 (5.145-1791.219) |
0.002 |
Data are shown as the number of patients (%)
aP-value calculated using the chi-square test or Fisher’s exact test
bP-value calculated using logistic regression for multivariate analysis
DDX11+, FPKM (fragments per kilobase of exon per million fragments mapped) > 20.0; TMEM38B-, FPKM < 5.0; PRUNE2-, FPKM < 32.0
Point 5: Fig.2: the PCA does not show a good classification of patients.
Response: We agree with the reviewer that PCA (Figure 2) does not show a good classification of patients in terms of aggressiveness since some of the aggressive ccRCC patients were grouped with non-aggressive ccRCC patients. However, this result showed that synchronous bone metastatic patients were grouped with non-aggressive ccRCC patients, while synchronous lung metastatic patients were grouped with the aggressive ccRCC group. Since aggressiveness of ccRCC that we defined in this study is not absolute terminology, PCA could not clearly classify patients into 2 groups. Moreover, this result demonstrated that the early bone metastasis group genetically clusters with the less aggressive ccRCC group which gives beneficial information to clinicians that prognosis of early bone metastasis ccRCC patients would not be bad. We have already discussed this information in the discussion section (lines 250-253) but edited in response to the reviewer’s comments (lines 253-255).
Point 6: SM Fig.5: The authors should also include RNA-seq data of the 6 mutated genes and then compare them with the TCGA data.
Response: Direct comparison of our RNA-seq data with the TCGA data was not possible since our dataset contains unique cases that were intentionally collected from aggressive cases of ccRCC. We acknowledge that limitation of our study is that we have not provided similar results from a different independent dataset. Future validation study is underway to overcome this limitation. We have added this information in the limitation section (lines 297-298).
Point 7: The authors did not explain and discuss why regulation of some genes (e.g., MOCOS, BAIAP2L1, NPR3) is not consistent with RNA-seq data.
Response: There has been not much information regarding some genes including the genes that reviewer has pointed out. We have reviewed every gene identified in this study and mentioned the most significant genes in relation to ccRCC according to previous studies. Genes that were not mentioned in this study were those that have no related studies of in relation to ccRCC. Moreover, due to the page limitation of the manuscript, we tried to reduce every unnecessary information as much as possible.
Point 8: SM Fig.6: How did they categorize the patients into low or high groups based on expression levels of the 16 genes?
Response: Patients were categorized into low or high groups by computing all possible cutoff values between the lower and upper quartiles, and the best performing threshold was used as a cutoff. We have added this information in the Supplementary Method 7.
Point 9: SM Table 4: why the expression level of DDX11 is smaller in samples with higher Fuhrman grade (3 and 4)?
Response: In Supplementary Table 4, we have reported that the expression level of DDX11 in higher Fuhrman grade (3 and 4) is 0.0148±0.0215 compared to 0.0059±0.0057 in lower Fuhrman grade (1 and 2), which shows that expressions level of DDX11 is higher in higher Fuhrman grade.
Point 10: How about the other 7 newly identified genes from RNA-seq data, I expect to see whether or not their regulation is consistent with the published qPCR results. Why the 6 previously identified genes (PBRM1, BAP1, SETD2, KDM5C, FOXC2, and CLIP4) do not show any power for predicting tumor aggressiveness?
Response: Thank you for your comments. We have reviewed every gene identified in this study and mentioned the most significant genes in relation to ccRCC according to previous studies. Other 7 newly identified genes were not mentioned in this study since there are no related studies. Therefore, whether the regulations of those genes are consistent or not could not be reported due to the lack of previous studies. Moreover, due to the page limitation of the manuscript, we tried to reduce every unnecessary information as much as possible.
The reason that 6 previously identified genes did not show any power for predicting tumor aggressiveness was small sample size. Moreover, inclusion of unique cases in this study might have resulted in exclusion of previously 6 identified genes, although they showed similar mutation frequency patterns as expected from other studies. This information has already been mentioned in the discussion section (lines 210-213), however in response to the reviewer’s comments, we have added the sentence (line 213).
Minor concerns
Point 11: In a recent publication (doi: 10.3390/cancers12010064), the SLC gene family genes were also identified as dysregulated in collecting duct renal cell carcinoma, which I think should be discussed by the authors.
Response: Thank you for your comments. The suggested article reported RNA sequencing of collecting duct carcinoma, a rare renal cell carcinoma subtype with a very poor prognosis. They reported that the most upregulated gene was keratin 17, and the most downregulated gene was cubilin. Solute carrier (SLC) gene family (SLC3A1, SLC9A3, SLC26A7, and SLC47A1) was included in the most downregulated genes. We included this information in the discussion section (lines 285-288).
Point 12: Tables: the tables should be re-formatted by adjusting the size of grids, making them easier to be read.
Response: We have readjusted the sizes of the grids and fonts of the tables in order to make them easier to be read.
Point 13: Figures: the resolution of most figures in supplementary materials is too low to be read.
Response: Thank you for your comments. We have inserted figures with low resolution since the file sizes were too big. However, in response to the reviewer’s comments, we have replaced the figures with higher resolution.
Point 14: Data sharing: the authors should consider making their RNA-seq data accessible by others. Either publish the read counts of genes as a supplementary file or submit the data to omnibus in Pubmed.
Response: Thank you for your comments. Data are not available to other researchers because they are from a registry of patients providing routinely collected data.

Round 2
Reviewer 2 Report
Table 2: The log2fc should have a +or- sign to indicate the direction of the gene regulation. Data sharing: i still think the authors should make the RNA-seq data publicly available, as it is important for other interested researchers to reuse the data. The patient's privacy can be hidden and protected by using some IDs that only indicate the types of cancer. A supplementary Excel file containing read counts or FPKM will be OK.Author Response
Point 1: Table 2: The log2fc should have a +or- sign to indicate the direction of the gene regulation.
Response: Thank you for your comments. We have indicated + and – sign to indicate the direction of the gene regulation.
Point 2: Data sharing: I still think the authors should make the RNA-seq data publicly available, as it is important for other interested researchers to reuse the data. The patient's privacy can be hidden and protected by using some IDs that only indicate the types of cancer. A supplementary Excel file containing read counts or FPKM will be OK.
Response: We have uploaded supplementary excel file containing read counts and FPKM. Thank you again for your effort and time reviewing our manuscript.
